# The Organization of the Golgi Structures during Drosophila Male Meiosis Requires the Citrate Lyase ATPCL

**DOI:** 10.3390/ijms22115745

**Published:** 2021-05-27

**Authors:** Patrizia Morciano, Maria Laura Di Giorgio, Liliana Tullo, Giovanni Cenci

**Affiliations:** 1INFN-Laboratori Nazionali del Gran Sasso, 67100 Assergi, Italy; 2Dipartimento di Biologia e Biotecnologie “Charles Darwin”, Sapienza–Università di Roma, 00185 Rome, Italy; marialauradg@gmail.com (M.L.D.G.); liliana.tullo@uniroma1.it (L.T.); 3Fondazione Cenci Bolognetti, Istituto Pasteur, 00185 Rome, Italy

**Keywords:** Drosophila, DmATPCL, Golgi stacks, spermatogenesis, Lava lamp, acroblast, meiosis

## Abstract

During spermatogenesis, the Golgi apparatus serves important roles including the formation of the acrosome, which is a sperm-specific organelle essential for fertilization. We have previously demonstrated that *D. melanogaster* ATP-dependent Citrate Lyase (ATPCL) is required for spindle organization, cytokinesis, and fusome assembly during male meiosis, mainly due to is activity on fatty acid biosynthesis. Here, we show that depletion of *DmATPCL* also affects the organization of acrosome and suggest a role for this enzyme in the assembly of Golgi-derived structures during Drosophila spermatogenesis.

## 1. Introduction

The metabolic ATP-dependent Citrate Lyase (ATPCL) enzyme converts mitochondria-exported citrate in oxaloacetate and high-energy metabolite acetyl-CoA, which is an intermediary for many biochemical reactions such as the synthesis of fatty acids, cholesterol and acetylcholine [1]. ATPCL is a high conserved enzyme and, along with its metabolites, plays an important role in many physiological (histone acetylation, gene regulation and DNA damage repair) [2,3] and pathological processes such as hyperlipidemia, hypercholesterolemia, diabetes type-2, and cancer [4,5,6,7]. We have recently shown that the Drosophila ATPCL (DmATPCL) is required for acetyl-CoA synthesis in somatic cells. However, although levels of acetyl CoA are reduced in *DmATPCL* mutants, unlike its human counterpart, the loss of DmATPCL does not affect global histone acetylation and gene expression [8]. Interestingly, we have also found that DmATPCL is required for a proper male meiosis as its depletion leads to defects in spindle organization, cytokinesis, and fusome assembly in both larval and adult testes [8]. The DmATPCL function in male meiosis is mainly attributable to its role in the biosynthesis of fatty acids but not to the reduction of acetylCoA synthesis that, similarly to mitosis, did not impact global lysine acetylation pattern in Drosophila male meiosis [8].

In fruit fly spermatogenesis, one stem cell, the gonialblast, undergoes four rounds of synchronous mitosis to produce a cyst of 16 primary spermatocytes, which grow 25 times in size. These cells are interconnected in a syncytium by intercellular bridges called ring canals and by a branched network, the fusome, an ER-derived germline-specific cytoskeleton. These structures allow communication, synchronization and differentiation of the germ cells [9,10]. At the end of the growth phase, the spermatocytes enter meiosis and differentiate into 64 haploid spermatids [11]. During spermatogenesis, the 64 round-shaped spermatids undergo maturation through a dramatic morphological change along with a substantial spatial rearrangement of the internal organelles, including the Golgi apparatus. Unlike the continuous membranous system called Golgi ribbon in mammals, the Drosophila Golgi apparatus consists of unconnected stacks that are dispersed throughout the cytoplasm in premeiotic cells. Just prior to and during the nuclear elongation phase of spermatogenesis, the Golgi stacks condense and assemble in a ribbon-like structure at the apical side of the spermatid nucleus to form the acroblast. This unique Golgi assemblage is essential for fertilization since it is required to organize the secretory pathway in the highly polarized spermatids [12,13] and for the subsequent acrosome formation, nuclear elongation and therefore sperm maturation. Despite several Golgi resident proteins having been detected and localized in Drosophila male meiosis, how Golgi-derived vesicles are organized throughout spermatogenesis has not been fully addressed. Here, we describe that DmATPCL localizes at the acroblast and that its loss impairs Golgi stack organization in primary spermatocytes and affects the acroblast structure in developing spermatids.

## 2. Results and Conclusions

### ATPCL Is Required for Acroblast Assembly and Golgi Integrity

We have previously characterized two male sterile *DmATPCL* mutant alleles (*DmATPCL^01466^* and *DmATPCL^DG23402^*) that displayed frequent multinucleated spermatids, irregular centrosome organization and spindle formation, as well as abnormal primary spermatocyte cysts during male meiosis indicating that an impairment of DmATPCL function affects male spermatogenesis at different levels [8]. To obtain more insights into the role of DmATPCL during male meiosis, we immunostained wild-type testes with our custom-made anti-ATPCL antibody [14]. We found no significant cellular localization during both meiotic divisions. However, in both onion stage and elongated spermatids we observed that DmATPCL showed a pronounced and specific juxtanuclear localization that was missing in *DmATPCL^DG23402^* homozygous and hemizygous mutants (we focused our analysis only on the hypomorphic *DmATPCL^DG23402^* allele, as the most severe *DmATPCL^01466^* mutant allele exhibited highly irregular postmeiotic figures that were not suitable for carrying out these observations [8] (Figure 1a). Interestingly, this localization pattern is similar to that described for the acroblast, a membranous structure situated at the anterior side of the elongating spermatids, which is composed of Golgi cisternae [15]. To confirm whether a functional relationship could exist between DmATPCL and the acroblast, we co-immunostained wild-type spermatids with anti-ATPCL and anti-Lava antibodies. We found that the antibody against the Golgi marker Lava (Lva), which normally stains the edge and the tip of the acroblast [12,16], partially surrounded the anti-ATPCL staining, confirming that DmATPCL co-localized with the acroblast (Figure 1b). Moreover, the anti-Lva immunostaining also revealed that, whereas in wild-type onion stage spermatids the Golgi aggregate close to the nucleus and form a ribbon-like structure that corresponds to the acroblast, a high proportion of *DmATPCL^DG23402^/Df(2R)Exel7138* mutant spermatids (74%; N = 320) exhibited dispersed Lva-containing vesicles instead (Figure 2m–o) indicating that loss of DmATPCL severely affected the assembly of the Golgi-based acroblast in spermatids. This observation prompted us to verify whether DmATPCL is also required to maintain a proper Golgi apparatus during spermatogenesis using anti-LVA antibody as a marker. With this aim, we performed a double immunostaining on both mutant and wild-type testes using the anti-LVA antibody, to stain the Golgi, and the anti-tubulin antibody, to clearly identify the different developmental stages and meiotic phases of the cells.

In wild-type Oregon R mature primary spermatocytes, Golgi structures appear as distinct ring or horseshoe-like structures (Figure 3a–c). At prometaphase I, the Golgi appear fragmented and this fragmentation is maintained through metaphase I, leading to the formation of LVA-positive spots that decrease in size and increase in number (Figure 2a). These observations are consistent with published results indicating that the Golgi apparatuses are disassembled during meiosis [13,16,17]. During anaphase I and telophase I, the Golgi are excluded from the spindle midzone, segregate to both poles and became enriched around the daughter nuclei (Figure 2a,b). After telophase I and in secondary spermatocytes, the Golgi are redistributed throughout the cytoplasm, but the cells lack the ring- and horseshoe-like structures typical of primary spermatocytes. The localization and distribution patterns of the Golgi during meiosis II is very similar to that of meiosis I, and ultimately they give rise to a ribbon-like formations in the onion stage spermatids as described above (Figure 2j). These ribbon-like formations break as the spermatids elongate leaving few acroblast remnants. The analysis of *DmATPCL^01466^/Df(2R)Exel7138* and *DmATPCL^DG23402^/Df(2R)Exel7138* mutant testes revealed that in pre-meiotic primary spermatocytes both number and size of Golgi stacks in mutant spermatocytes were different compared to control. In particular, the Golgi appeared fragmented and consequently LVA-positive spots more numerous and smaller than control (~40–50 vs. ~20 stacks/cells; *p* < 0.05) (Figure 2d,g,j,k), which is also reminiscent of the dispersed Golgi structures in the onion stage spermatids. Furthermore, the Golgi distribution and segregation during both normal and irregular meiotic divisions (appreciable only in the DmATPCL^DG23402^ mutant combinations; see above) appeared normal, indicating the ATPCL is not required for the Golgi distribution (Figure 2g). However, whether the effects on the Golgi specifically influence the entry (cis) face, where newly synthesized proteins from the ER enter the Golgi, or the exit (trans face), where they leave the Golgi, remains unaddressed. Overall, our observations indicate that in the absence of DmATPCL, meiotic cells fail to properly assemble the Golgi, suggesting that DmATPCL regulates the dynamic nature of the Golgi apparatus. Defects in the Golgi organization were rescued by the expression of a wild-type *UAS DmATPCL* transgene under the control of a *TubGal4* promoter in *DmATPC^LDG23042^* mutant background, confirming that they indeed arose as a consequence lesions in the *DmATPCL* gene. Yet, this requirement of DmATPCL seems restricted only to spermatogenesis as the Golgi organization in somatic tissues of *DmATPCL* mutant larvae appeared normal (Appendix A), confirming a predominant role for DmATPCL specifically during male-meiosis [8,14].

Given the multiple roles played by ATPCL, it remains arduous to identify a single cause to explain the Golgi phenotype, including the loss of acroblast of *DmATPCL* mutants. Despite our previous finding that acetyl-CoA levels were reduced in *DmATPCL* mutant testes and that this reduction had no effect on global protein acetylation [8], we cannot rule out the possibility that the loss of specific proteins acetylation of Golgi proteins could be taken into account to explain the phenotype described. Moreover, an impairment of cytosol-to-ER acetyl-CoA flux has been recently shown to affect the Golgi apparatus in mouse cells [18], thus linking that the ATPCL requirement for a proper Golgi to its capacity to generate the high-energy metabolite acetyl-CoA. The growing number of papers showing that the assembly of the Golgi in different organisms, including plants, is dependent on actin cytoskeleton (reviewed [19,20,21]) could also provide an additional explanation for the defective Golgi organization. Indeed, our previous results have indicated that loss of DmATPCL influences fusome branching, very likely by affecting F-actin assembly [8]. As the disruption of actin machinery is known to alter the pairing of the Golgi stacks in both Drosophila S2 and human cells [22], it can be argued the that defects in Golgi organization in *DmATPCL* mutant testes mainly result from perturbation of F-actin assembly. Interestingly, a genome-wide RNA-mediated interference screen in S2 cells revealed that genes involved in fatty acid biosynthesis are also required for the Golgi organization [23]. Thus, since depletion of ATPCL reduces fatty acid levels, it can be envisaged that defects in the assembly of Golgi-derived structures in *DmATPCL* mutants could also arise as a consequence of low levels of fatty acids. It can be also speculated that the ATPCL-derived lipids could play a direct role in the acroblast, thus explaining the specific enrichment of this enzyme in the acroblast. The hydrolysis of sphyngomyelins to ceramides, mostly affecting species with very long chain polyenoic fatty acids, has been demonstrated to occur in rat spermatozoa during capacitation and acrosome reaction [24]. We can thus hypothesize that, although sperm capacitation has not been completely addressed in Drosophila, the acroblast-based ATPCL localization could serve to supply Drosophila spermatozoa with the lipid classes required for acrosome. It is also worth noting that DmATPCL enrichment in Drosophila spermatids is consistent with previous observations showing increased ATP citrate lyase and acetyl-CoA carboxylase activities found in post-meiotic germ cells (spermatids) of adult rats [25]. From this perspective, the localization of ATPCL in spermatids, similarly to its rat counterpart, it is necessary for ensuring high rate of synthesis of cell-specific lipids in this cell type.

Finally, although the four phases formation of the acrosome (Golgi phase, cap phase, acrosome phase and mature phase) has been characterized in human, the molecular mechanisms underlying this process, as well as how defects in acrosome biogenesis leads to infertility [26], are still largely unknown. Thus, our study could provide a cue to investigate the role of the human counterpart ACL in acrosome formation and its potential involvement in human infertility.

## 3. Materials and Methods

### 3.1. Drosophila Strains and Crosses

The insertion lines *DmATPCL^01466^* and *DmATPCL^DG23402^*, as well as the *Df(2R)Exel7138* that uncovers *DmATPCL*, were obtained from the Bloomington Stock Center and were balanced over *CyTb*, as previously described [8,14]. *Oregon-R* flies were used as wild-type control. Flies were raised on standard corn-meal food and maintained under a 12 h light/dark cycle.

### 3.2. Cromosome Cytology, Immunostaining, and Microscopy

Fixation and staining were performed on dissected third-instar larvae testes as previously described [8]. The primary antibodies and the dilutions used were as follows: Anti-tub (1:1000) (Sigma-Aldrich, St. Louis, MO, USA), anti-DmATPCL (1:100) [14], anti-Lava lamp (1:300) [16]. The secondary antibody incubation was performed using FITC-conjugated anti-mouse IgG + IgM (1:20; Jackson ImmunoResearch Laboratories, Cambridge, UK), Alexa 555-conjugated anti-rabbit IgG (1:300 in PBS; Molecular Probes, Eugene, OR, USA) and Alexa Fluor 488-conjugated anti-guinea pig IgG (1:300 in PBS; Jackson laboratories) for 2 h at room temperature. Slides were then mounted in Vectashield medium H-1200 with DAPI (Vector Laboratories, Burlingame, CA, USA) to stain DNA and reduce fluorescence fading. Slides were analyzed using a Zeiss Axioplan epifluorescence microscope (Carl Zeiss, Obezkochen, Germany), equipped with a cooled CCD camera (Photometrics, Woburn, MA, USA). Gray-scale digital images were collected separately, converted to Photoshop format, pseudo-colored, and merged. Quantification of Lva spot was carried out using the ImageJ software. Box plots were obtained with Python 3.3.

## Figures and Tables

**Figure 1 ijms-22-05745-f001:**
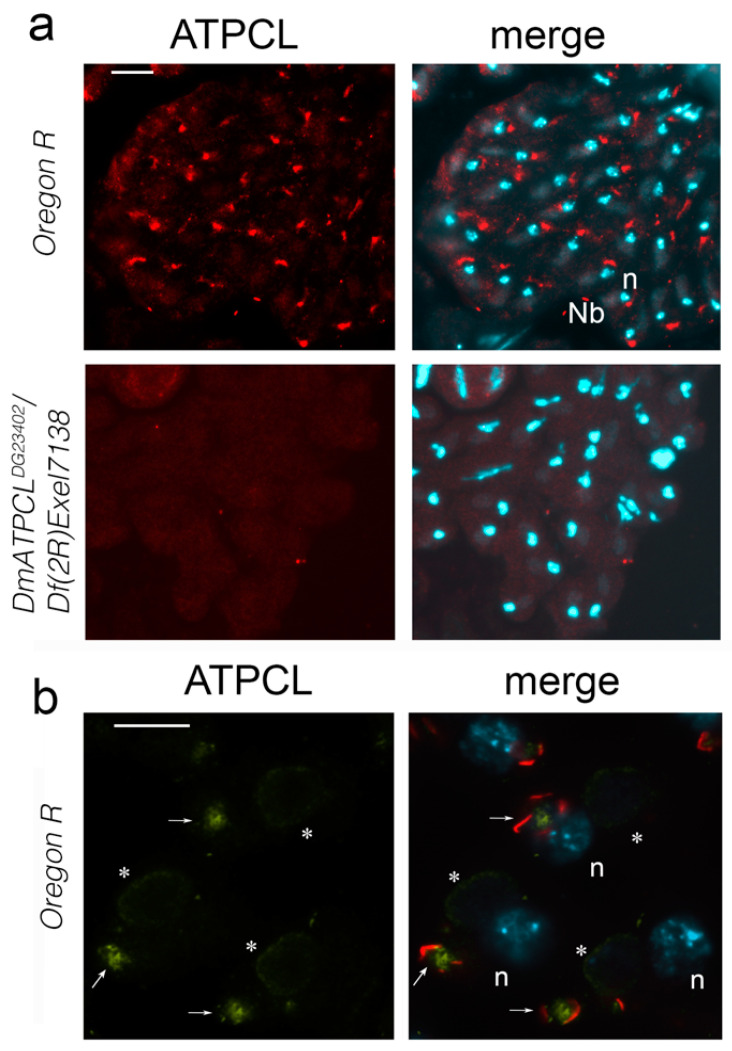
ATPCL localizes at the acroblast. (**a**) Partial cysts of spermatids from wild-type (*Oregon R*) and *DmATPCL^DG23402^/Df(2R)Exel7138* hemizygous mutants stained for ATPCL (red) and DAPI (blue). Note a juxtanuclear localization of ATPCL that is missing in the mutant spermatids. Localization of ATPCL in the nebenkern (Nb) is unspecific. Scale bar = 10 µm; (**b**) Wild-type onion stage spermatids stained for ATPCL (green in the merge), the Golgi marker Lava Lamp (LAVA, red) and DAPI (blue) showing that ATPCL colocalizes with LAVA in the acroblasts (arrows). Nb = Nebenkern; *n* = haploid spermatid nuclei; * = unspecific staining on nebenkern. Scale bar = 5 µm.

**Figure 2 ijms-22-05745-f002:**
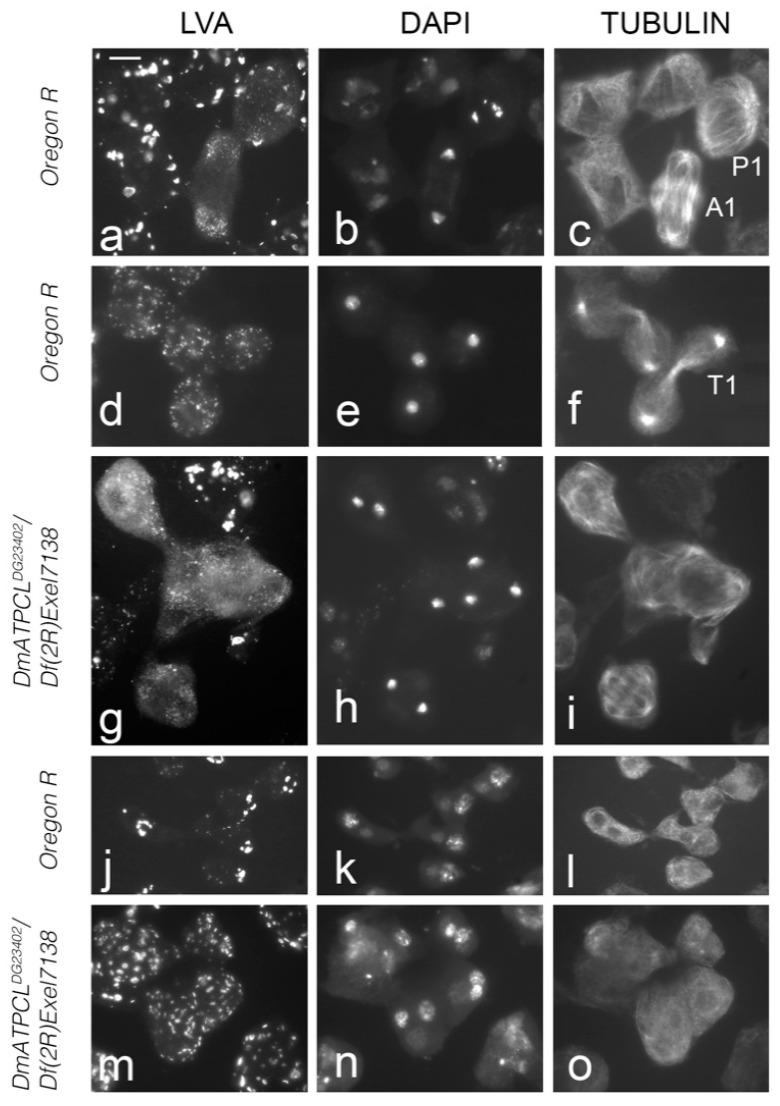
Loss of DmATPCL alters the organization, but not the segregation, of the Golgi structures, during male meiosis. The localization pattern of the Golgi in wild-type (**a**–**f**,**j**–**l**) and mutant (**g**–**i**,**m**–**o**) male meiotic divisions is similar. Note that *DmATPCL* mutant cell divisions are characterized by multipolar spindles, as previously described [8]. See text for the description of the Golgi localization pattern. Note that whereas in wild-type onion stage spermatids the Golgi form a well-defined structure, the acroblast (**j**–**l**), in *DmATPCL* mutants, which also exhibit irregular nebenkern-nucleus associations, the Golgi appear fragmented (**m**–**o**) leading to the loss of the acroblast. P1 = Prometaphase I; A1 = Anaphase I; T1 = Telophase I. Scale bar = 10 µm.

**Figure 3 ijms-22-05745-f003:**
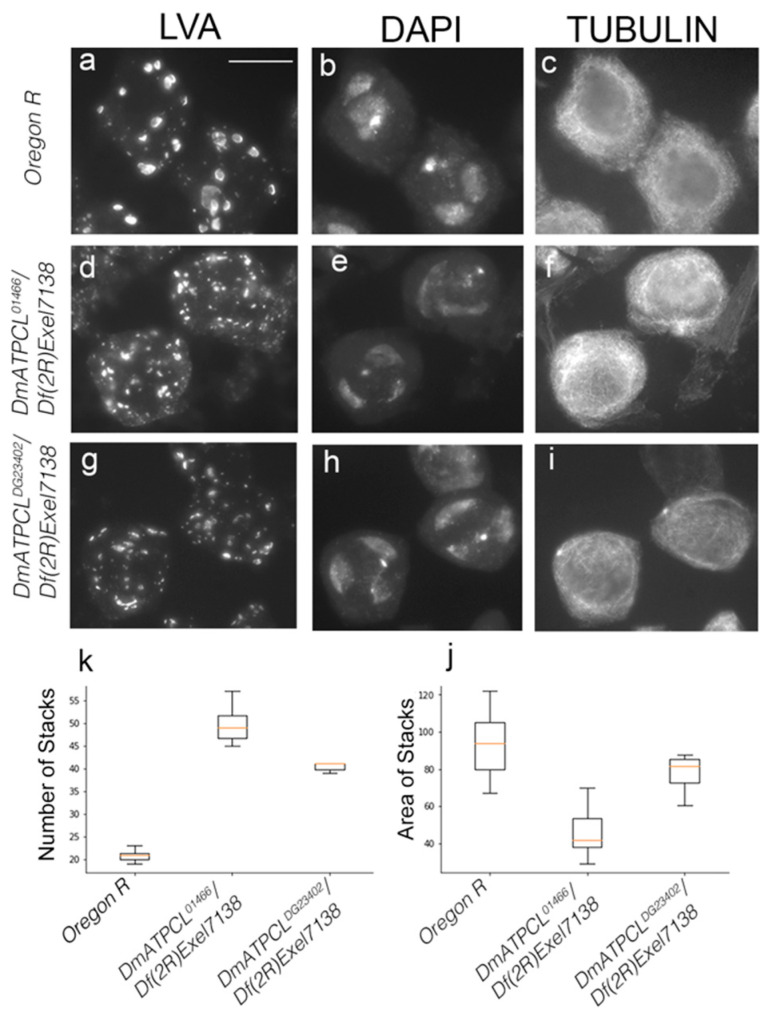
Depletion of DmATPCL affects the Golgi apparatuses in primary spermatocytes. (**a**) Primary spermatocytes from wild-type *Oregon R* (**a**–**c**) and mutant (**d**–**i**) testes stained for Lava (Lva; **a**,**d**,**g**), DAPI (**b**,**e**,**h**) and tubulin (**c**,**f**,**i**). Note that the Golgi stacks organization, as revealed by the anti-Lva immunostaining, is altered in *DmATPCL* mutant hemizygous combinations. Tubulin staining indicates that all primary spermatocytes are at the same developmental stage (S4–S5) [11]. (**b**) Box plots showing the quantification of number and extension of Lva-positive Golgi structures from 20 wild-type and mutant primary spermatocytes. See text for further details (*p* < 0.05, t-Student Test). Scale bar = 10 µm.

## Data Availability

All data generated or analyzed during this study are included in this published article and its Appendix A.

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
