# Peer review of "The Organization of the Golgi Structures during Drosophila Male Meiosis Requires the Citrate Lyase ATPCL"

_ijms, 2021, doi:10.3390/ijms22115745_

Round 1
Reviewer 1 Report
To Authors:
Dear Authors,
The article " The organization of the Golgi structures during Drosophila male meiosis requires the Citrate Lyase ATPCL" by P. Morciano et al., aimed to characterize Citrate Lyase ATPCL as a key player in during Drosophila spermatogenesis and depletion of DmATPCL affects also the organization of acrosome and suggest a role for this enzyme in the assembly of Golgi-derived structures.
The work performed in this paper used Drosophila Strains. Experiments and assays used in the current study are up-to-date and well established. The study is not limited by Cromosome Cytology, Immunostaining, and Microscopy experiments and is supplemented by animal studies that add significant value to this work.
Main concerns and comments:
Question 1. Is the whole Golgi structure affected or phenotype is limited by trans-side only? Despite Figure 1 demonstrating IF data for cells with or without Depletion of DmATPCL in primary spermatocytes is very interesting, I would suggest to reinforce it by adding additional Golgi markers (cis- and trans-parts) and show colocalization (for example dGM130 as early Golgi ). And probably supplement with quantification if there is change in colocalization events in WT vs mATPCLDG23402/Df(2R)Exel7138 samples.
Minor issue for same figure: line 158 (Figure legend): Scale bar= 10?m;
Question 2. I would suggest bringing images on Figure 2 to the same size.
Line 183: A1= anaphase I; T1= Telophase I; Scale bar = 10μm
Author Response
Please see the attachment (revised manuscript).
Dear Editor,
we thank both reviewers for their comments and suggestions.
Please enclosed find a new version of our manuscript that has been revised according to the referees’ suggestions. In this new version, we have addressed all reviewers’ concerns (as you can see also from the attached point-by-point response). In the revised version we have highlighted in blue all changes made in both the text and figure legends as requested by the reviewers.
We really hope that you will find this new version of our manuscript acceptable in this form for publication in IJMS
Sincerely yours,
Patrizia Morciano
Giovanni Cenci
Response to REVIEWER 1
The article " The organization of the Golgi structures during Drosophila male meiosis requires the Citrate Lyase ATPCL" by P. Morciano et al., aimed to characterize Citrate Lyase ATPCL as a key player in during Drosophila spermatogenesis and depletion of DmATPCL affects also the organization of acrosome and suggest a role for this enzyme in the assembly of Golgi-derived structures.
The work performed in this paper used Drosophila Strains. Experiments and assays used in the current study are up-to-date and well established. The study is not limited by Cromosome Cytology, Immunostaining, and Microscopy experiments and is supplemented by animal studies that add significant value to this work.
We thank this reviewer for her/his positive feedback.
Main concerns and comments:
Question 1. Is the whole Golgi structure affected or phenotype is limited by trans-side only? Despite Figure 1 demonstrating IF data for cells with or without Depletion of DmATPCL in primary spermatocytes is very interesting, I would suggest to reinforce it by adding additional Golgi markers (cis- and trans-parts) and show colocalization (for example dGM130 as early Golgi ). And probably supplement with quantification if there is change in colocalization events in WT vs mATPCLDG23402/Df(2R)Exel7138 samples.
We agree with the reviewer that it will be interesting to check whether the effects on the Golgi involve the whole structures or are limited to either the trans- or cis-side. Although we think that this detailed characterization is a little beyond the scope of this short communication, we plan in the future to employ cis-side markers (not yet available in the lab and likely difficult to obtain in the time frame required for this revision) to address this point. Nevertheless, I would like to point out that we had already tried to study the localization pattern of dGM130 in both wild-type and mutant testes. However, we were not able to get the commercial anti-dGM130 antibody (ab30637) to work on male meiotic tissues even by using different fixation techniques. Scarce results with the same abcam lot were also obtained by other groups (M.G. Giansanti and A. Frappaolo personal communication) confirming that this antibody is not a reliable reagent.
We have performed a preliminary comparison of lava lamp spots intensities among control and dmATPCL mutants and found that total Lva intensity of Golgi fragments in the mutant spermatocytes is equivalent of that observed in the wild-type intact Golgi stacks suggesting that the fragmentation influences the entirety of the Golgi. Yet, these data do not allow to conclude whether there are different effects on the cis-side structures compared to trans-side Golgi sections. This has been indicated in the revised version on line 115.
“However, whether the effects on the Golgi influences specifically the entry (cis) face, where newly synthesized proteins from the ER enter the Golgi, or the exit (trans face), where they leave the Golgi, it remains unaddressed”.
Minor issue for same figure: line 158 (Figure legend): Scale bar= 10µm;
We have addressed this criticism in the revised Figure legend.
Question 2. I would suggest bringing images on Figure 2 to the same size.
The previous original 2 has been replaced by a new figure 2 with panels of the same size
Line 183: A1= anaphase I; T1= Telophase I; Scale bar = 10μm
We have addressed this criticism in the revised Figure legend.
Reviewer 2 Report
Nice study. I'm missing few minor explanations. Add 1-2 sentences why acrosome/acroblast is so important for fertilization. I also didn't see explanation why you used tubulin for staining? I would also suggest to add few sentences what your data could possible mean for humans.
Author Response
Please see the attachment (revised manuscript).
Dear Editor,
we thank both reviewers for their comments and suggestions.
Please enclosed find a new version of our manuscript that has been revised according to the referees’ suggestions. In this new version, we have addressed all reviewers’ concerns (as you can see also from the attached point-by-point response). In the revised version we have highlighted in blue all changes made in both the text and figure legends as requested by the reviewers.
We really hope that you will find this new version of our manuscript acceptable in this form for publication in IJMS
Sincerely yours,
Patrizia Morciano
Giovanni Cenci
Response to REVIEWER 2
Nice study. I'm missing few minor explanations.
We thank this reviewer for her/his positive comments.
Add 1-2 sentences why acrosome/acroblast is so important for fertilization.
We have added the following sentence (on line 52).
“This unique Golgi assemblage is essential for fertilization since it is required to organize the secretory pathway in the highly polarized spermatids [12,13] and for the subsequent acrosome formation, nuclear elongation and therefore sperm maturation”
I also didn't see explanation why you used tubulin for staining?
We added the following sentence to support the use of the anti-tubulin antibody (on line 89).
“To this aim, we performed a double immunostaining on both mutant and wild-type testes using the anti-LVA antibody, as a marker for the Golgi, and the anti-tubulin antibody, to clearly identify the different developmental stages and meiotic phases of the cells.
I would also suggest to add few sentences what your data could possible mean for humans.
We thank the reviewer for this suggestion. We have added a final sentence (line 160) about a potential implication of ACL in human fertility.
“Finally, although the four phases-formation of the acrosome (Golgi phase, cap phase, acrosome phase and mature phase) has been characterized in human, the molecular mechanisms underlying this process as well as how defects in acrosome biogenesis leads to infertility [26], are still largely unknown. Thus, our study could provide a cue to investigate the role of the human counterpart ACL in acrosome formation and its potential involvement in human infertility”.